# Interferon Regulator Factor 5: A Novel Inflammatory Marker and Promising Therapeutic Target in Ulcerative Colitis

**DOI:** 10.3390/biomedicines13092251

**Published:** 2025-09-12

**Authors:** Karima Farrag, Aysegül Aksan, Marina Korotkova, Helena Idborg, Per-Johan Jakobsson, Andreas Weigert, Michael Vieth, Stefan Zeuzem, Irina Blumenstein, Jürgen Stein

**Affiliations:** 1Medical Clinic 1, University Hospital, Goethe University Frankfurt, 60594 Frankfurt am Main, Germanyblumenstein@em.uni-frankfurt.de (I.B.); 2Interdisciplinary Crohn Colitis Centre Rhein-Main, 60594 Frankfurt am Main, Germanyj.stein@em.uni-frankfurt.de (J.S.); 3Division of Rheumatology, Department of Medicine, Solna, Karolinska Institute, Karolinska University Hospital, SE-171 76 Stockholm, Sweden; 4Center for Molecular Medicine, Karolinska University Hospital, SE-171 76 Stockholm, Sweden; 5Department of Pharmaceutical Chemistry, Goethe University of Frankfurt, 60594 Frankfurt am Main, Germany; 6Institute for Pathology, Klinikum Bayreuth, 95445 Bayreuth, Germany; 7Department of Gastroenterology and Clinical Nutrition, DGD Clinics Sachsenhausen, Teaching Hospital of the Goethe University, 60594 Frankfurt am Main, Germany

**Keywords:** inflammatory bowel disease, ulcerative colitis, IRF5, immunohistochemistry

## Abstract

**Background:** Inflammatory bowel disease (IBD), including Crohn’s disease (CD) and ulcerative colitis (UC), is characterized by chronic inflammation affecting the gastrointestinal tract and extraintestinal organs. The etiology of IBD is multifactorial, involving genetic, immunological, and environmental factors. Over 200 genetic loci have been associated with the disease, indicating a significant genetic predisposition. Despite advances in understanding its genetic basis, clinical management remains challenging due to heterogeneity in disease presentation and variable treatment responses. Current therapies, such as 5-aminosalicylates and biologics, are not universally effective, underscoring the need for reliable biomarkers to predict therapeutic responses. **Objective:** This study investigates the potential role of interferon regulatory factor 5 (IRF5) in the pathogenesis of IBD, with a particular focus on UC. **Methods:** We conducted a systematic analysis of colon biopsies from 30 adult patients diagnosed with UC and from 8 non-IBD controls. Immunostaining was performed to assess IRF5 expression in colonic tissues using the primary IRF5 antibody (1:300, Abcam, ab181553). Statistical analyses evaluated the correlation between IRF5-positive cell counts, disease activity, and inflammatory markers such as calprotectin. **Results:** Our analysis revealed a significant increase in IRF5-positive macrophage-like cells in the inflamed mucosa of IBD patients compared to healthy controls. The number of IRF5-positive cells showed a positive correlation with disease activity and calprotectin levels, indicating that higher IRF5 expression is associated with increased inflammation. **Conclusions:** This study demonstrates a significant correlation between IRF5 expression and disease activity in UC, suggesting that IRF5 may play a crucial role in the inflammatory processes of the disease. The findings propose IRF5 as a novel biomarker for therapeutic intervention in IBD. Further research is needed to clarify the mechanisms by which IRF5 contributes to IBD pathogenesis and to explore the therapeutic potential of targeting this pathway in clinical settings.

## 1. Introduction

Inflammatory bowel disease (IBD) encompasses chronic inflammatory conditions such as Crohn’s disease (CD) and ulcerative colitis (UC) [1,2]. These disorders are characterized by inflammation not only in the gastrointestinal tract but also in extraintestinal organs, resulting in a broad spectrum of clinical manifestations. The etiology of IBD is complex, involving a combination of genetic, immunological, and environmental factors [3]. While the precise pathogenesis remains unclear, IBD is widely regarded as an immunodysregulatory disease, with a loss of immune homeostasis contributing to chronic inflammation [4].

A significant body of evidence supports the role of genetic factors in IBD. Genome-wide association studies (GWAS) and candidate gene studies have identified over 200 loci associated with IBD, including 99 loci shared between CD and UC [5,6,7]. These findings highlight the critical role of genetic predisposition in disease development. Familial clustering of IBD and higher concordance rates among monozygotic twins compared to dizygotic twins further underscore the genetic contribution [3,8]. Many of the implicated genes are involved in immune regulation and response, linking genetic variations to immune dysregulation and the overproduction of pro-inflammatory cytokines [6,7,9]. This connection not only highlights IBD pathogenesis but also suggests an overlap with other autoimmune diseases, which often share similar genetic backgrounds.

Despite advances in understanding the genetic basis of IBD, clinical treatment remains challenging due to the diverse clinical presentations of the patients. Furthermore, while current therapies—including 5-aminosalicylates, immunosuppressants, biologics, and nutritional interventions—are effective for many patients, they are not universally successful. Biomarkers that can predict therapeutic response or stratify patients based on disease subtypes are urgently needed. However, most candidate biomarkers lack validation in large cohorts or defined biological significance, limiting their clinical utility.

In this context, interferon regulatory factor 5 (IRF5) has emerged as a promising candidate for further investigation. The IRF5 gene, located on chromosome 7q32, encodes a transcription factor involved in immune regulation, particularly in the regulation of macrophages. Genetic variants in IRF5 that reduce its expression are associated with decreased susceptibility to UC [10] and other immune-mediated diseases, such as systemic lupus erythematosus (SLE) [11,12], rheumatoid arthritis, and primary biliary cirrhosis [13,14].

The role of the Type 1 IFN system in SLE has been thoroughly described [15], and plasmacytoid dendritic cells are known to be key interferon producers. IRF5 is present in the circulation of patients with SLE [12], and autoantibodies against IRF5 have recently been characterized [16].

The interferon (IFN) regulatory factor (IRF) family comprises nine transcription factors involved in antimicrobial defense, cellular survival, and hematopoietic development.

Some IRF members are activated in response to infections and contribute to innate immunity. IRF3 and IRF7 act as direct transducers of virus-mediated signaling, whereas IRF5 is implicated in responses to viral infections and Toll-like receptor (TLR) signaling [13,17,18,19].

IRF5 plays a key role in the differentiation of macrophages into a pro-inflammatory phenotype. Upon activation, it induces the expression of pro-inflammatory cytokines such as interleukin (IL)-6, IL-12, IL-23, and tumor necrosis factor-α (TNF-α), all of which are central to the inflammatory cascade in inflammatory bowel disease (IBD) [18,20,21,22,23,24,25,26]. These observations suggest a potential role for IRF5 in modulating intestinal immune responses and driving inflammation in IBD.

IRF5 is constitutively expressed in the cytoplasm of plasmacytoid dendritic cells, B cells, monocytes, and macrophages and is induced upon activation of TLR3, TLR4, TLR7/8, and TLR9 pathways in most lymphocytes [20,21].

IRF5 is activated following engagement of pathogen recognition receptors (PRRs) and downstream effector molecules. Chang et al. identified the nucleotide-binding oligomerization domain-containing protein 2 (NOD2) as a significant activator of IRF5 [27].

Upon recognition of microbial ligands, TLRs recruit the Toll/interleukin-1 receptor (TIR) domain-containing adaptor protein MyD88, which initiates signal transduction. MyD88 assembles into an oligomeric scaffold—the myddosome—together with interleukin-1 receptor-associated kinase 4 (IRAK4) and IRAK1 or IRAK2.

IRF5 interacts directly with the intermediary and TIR domains of MyD88 and is activated by post-translational modifications (PTMs), such as polyubiquitination by tumor necrosis factor receptor-associated factor 6 (TRAF6) and E3K63 ubiquitin ligase, as well as phosphorylation by IκB kinase β (IKKβ) and TANK-binding kinase-1 (TBK1) or receptor-interacting protein 2 (RIP2) [28].

RIP2, which functions downstream of NOD2, has been shown to be a potent activator of IRF5-mediated gene expression.

Protein-tyrosine kinase 2-beta (PTK2B/PYK2) is another putative IRF5 kinase involved in TLR signaling. It interacts with IRF5, promoting its phosphorylation, nuclear translocation, and DNA binding at specific gene promoters [17,27,29,30].

Activated IRF5 forms homodimers, translocates to the nucleus, and binds to cis-regulatory elements of genes encoding type I IFNs (IFN-α and IFN-β) and pro-inflammatory cytokines, thereby inducing their expression [26].

The specific effects of IRF5 mutations in IBD patients have not been thoroughly investigated to date. Based on the effects of gain-of-function mutations in other autoimmune diseases and the effects of mutations in the IL-10, IL-23, and NOD2 pathways, it appears likely that increased IRF5 expression will affect the response to TLR and Nod-like receptor sensing of bacterial antigens [14].

Increased signaling via TLRs and Nod-like receptors leads to a higher expression of pro-inflammatory cytokines such as TNF and decreased expression of IL-10, which could combine to promote a pro-inflammatory environment and contribute to intestinal pathology [31,32].

These findings suggest that IRF5 plays a central role in immune homeostasis and may act as a nodal point in the pathogenesis of multiple inflammatory diseases. However, its specific role in intestinal inflammation and IBD remains poorly understood [20,28,33].

Currently, the treatment for IBD mainly includes 5-amiosalicylates, immunosuppressants, biologics, and nutritional therapy. The therapeutic effect of IBD is evaluated by clinical, endoscopic, and histological criteria. To date, there are no specific molecular markers to evaluate therapeutic efficacy. Novel biomarkers are urgently needed to define possible subgroups of IBD in clinical trials and to classify patients with IBD according to their likelihood of treatment response to specific drugs in clinical practice [1,2]. While several biomarker studies have been performed, few have been validated in a larger cohort or assigned biological significance. The characterization of biomarkers might offer insights into disease pathogenesis in different subgroups of patients and could be used to suggest novel therapeutic targets.

Protein-tyrosine kinase 2-beta (PTK2B/PYK2) is a putative IRF5 kinase. PYK2-deficient macrophages display impaired endogenous IRF5 activation, leading to a reduction in inflammatory gene expression [29]. Meanwhile, a PYK2 inhibitor, defactinib, has a similar effect on IRF5 activation in vitro and induces a transcriptomic signature in macrophages similar to that caused by IRF5 deficiency [34].

The aim of this study is to evaluate the involvement of IRF5 in the pathogenesis of IBD through a systematic analysis of IRF5-positive cells in colon biopsies from IBD patients. By investigating the expression and functional implications of IRF5 in intestinal inflammation, this study seeks to advance our understanding of IBD pathogenesis and identify potential therapeutic targets.

## 2. Materials and Methods

### 2.1. Ethics Statement

This study was conducted as an experimental study according to the ethical principles of the Declaration of Helsinki in adult patients previously diagnosed with IBD and non-IBD controls. The study design was approved by the local ethics committee, the Landesärztekammer Hessen (FF40/2017).

### 2.2. Study Population

Systematic biopsies from the various sections of the colon from eligible patients from the Interdisciplinary Crohn Colitis Centre (ICCC) Rhein-Main (Frankfurt am Main, Germany) who had been diagnosed with UC according to standard clinical, radiological, and pathological criteria were analyzed. The control group consisted of adults without IBD attending the same center for routine examination. Inclusion criteria were age of 18–65 years and body mass index (BMI) < 30 kg/m^2^. Exclusion criteria were malignant underlying disease, severe metabolic disease, cardiac, pulmonary, renal, and hepatic dysfunction or disease, and substance abuse (alcohol and drugs).

We collected biopsies from 8 healthy controls (HC), 30 UC patients in remission and an inflammatory status of disease. Two samples were taken from each intestinal section and stored directly on dry ice (caecum, ascending colon, colon transversum, descending colon, sigmoid colon, rectum). Additionally, we retrospectively looked for the routinely determined fecal calprotectin values at the time the biopsies were taken.

### 2.3. Immunostaining Analysis

We performed immunostaining analyses of the biopsies at the Karolinska institutet, Stockholm, and the clinic for pathology, Bayreuth.

For IRF5 staining, first, the samples were cut into tissue sections at 7 μm thickness and mounted onto slides. The tissue sections were fixed in formalin for 5 min at room temperature and were rinsed thoroughly with water to remove excess formalin. The sections were incubated with Fast-Enzymes (protease) for 10 min at room temperature to facilitate antigen retrieval. Then, the slides were washed three times for 3 min each with wash buffer to remove residual enzyme. Afterwards, the slides were incubated with 3% H_2_O_2_ in distilled water for 10 min at room temperature to quench endogenous peroxidase activity. After washing once with wash buffer for 2 min, the slides were incubated with Blocking Kit I for 10 min at room temperature to prevent non-specific antibody binding and washed again with wash buffer for 2 min. The duration for incubating the slides with the primary antibody (1:300, Abcam (Cambridge, UK), ab181553) was 60 min at room temperature. Then, the slides were washed again three times for 5 min each with wash buffer and then incubated with Post Block solution (including normal serum) for 30 min at room temperature to reduce background. After washing three times for 5 min each with wash buffer, the incubation with the secondary antibody (HRP-polymer and Avidin/Biotinylated enzyme peroxidase complex) was followed for 30 min at room temperature. Another cycle of washing, three times for 2 min each with wash buffer, was performed. The slides were incubated with Chromogen Development (DAB) substrate for 5–15 min, monitoring the color development until the desired intensity was achieved, and then they were rinsed thoroughly with distilled water to stop the DAB reaction.

The sections of one biopsy of each patient’s intestinal section were evaluated. The cells were counted using the InForm^®^ Software v2.4.10 program.

### 2.4. Statistical Analysis

Statistical analyses were conducted using IBM SPSS Statistics Version 29.0.2.0 (International Business Machine Corporation, Endicott, NY, USA).

We performed linear regression with mixed effects, with the number of IRF5-positive cells as the dependent variable, the patient as the random effect (there are several measurements per patient), and the inflammation degree and the colon region as fixed effects. Age and sex are considered as confounders.

We performed a linear regression for the dependent variable, the maximal number of IRF5-positive cells, with the predictor calprotectin. Age and sex were considered as confounders.

## 3. Results

Colon biopsies from 30 patients with UC (11 m, 36.67%, mean age of 42.7 years) were collected. Of those 30 patients, 9 patients were in clinical and endoscopic remission (Table 1).

Additionally, we collected biopsies from eight healthy control patients (4 m, 50%, mean age 48 years) without a history of UC or other intestinal disease and with no macroscopic or microscopic signs of inflammation.

We found that IRF5 was mainly expressed on macrophage-like cells by immunohistochemical staining (Figure 1). The proportion of IRF5+ cells was obviously much higher in intestinal tissues from IBD patients than that from HCs.

The degree of inflammation is significantly associated with the number of IRF5-positive cells (number of positively stained cells per tissue area in percent) (*p* < 0.001). There is a positive association (regression coefficient = 23.97): the average number of cells increases by 23.97 if the inflammation degree goes up by 1 (Figure 2).

A significant positive association between the maximal number of IRF5-positive cells per patient and the marker calprotectin was identified (*p* = 0.021). When calprotectin increases by 100 units, an average increase in the maximal number of IRF5-positive cells by 4.8 is estimated (Figure 3).

## 4. Discussion

Ulcerative colitis (UC), a subtype of inflammatory bowel disease (IBD), is characterized by chronic inflammation of the colonic mucosa, which can lead to significant morbidity and a reduced quality of life for affected individuals. The pathogenesis of UC is multifaceted, involving an abnormal immune response to the intestinal microbiota that results in an imbalance between pro-inflammatory and anti-inflammatory pathways [1,2,35,36]. Recent studies have underscored the potential role of Interferon Regulatory Factor 5 (IRF5) in this context.

IRF5 has been implicated in various autoimmune diseases, including multiple sclerosis and systemic lupus erythematosus (SLE) [37]. However, its specific role in IBD, particularly UC, remains to be fully elucidated. Our investigation into IRF5 expression in colonic mucosal tissues revealed a notable elevation in inflamed mucosa from IBD patients, suggesting that IRF5 may play a crucial role in the inflammatory processes associated with UC.

Despite extensive genome-wide association studies identifying over 200 loci associated with IBD, such as NOD2, ATG16L1, and IL-23R, these genetic variants account for only a fraction of the heritability observed in Crohn’s disease (CD) and UC [6].

This gap highlights the necessity for further research to identify additional pathogenic genes that contribute to the complexity of these diseases. Our findings indicate that increased IRF5 expression correlates with disease activity in UC, positioning it as a potential target for therapeutic intervention [33,38].

Moreover, IRF5’s role in modulating immune responses, particularly in B cells, dendritic cells, and macrophages, emphasizes its importance in the inflammatory cascade. IRF5 has been shown to play critical roles in immune responses and is involved in the pathogenesis of several immune diseases, including SLE, Sjögren’s syndrome, and psoriasis [17,20,21,31].

IRF5 plays a key role in the inflammatory response and is believed to contribute to the pathogenesis of ulcerative colitis (UC) through multiple mechanisms. Notably, it promotes the activation of macrophages and dendritic cells, driving their polarization toward a pro-inflammatory M1 phenotype and enhancing the production of cytokines such as TNF-α, IL-6, IL-12, and IL-23. These cytokines amplify intestinal inflammation and help sustain a cycle of immune activation [17,21,31].

IRF5 is also activated downstream of Toll-like receptor (TLR) signaling in response to microbial components in the gut, thereby enhancing innate immune responses and contributing to chronic inflammation in UC. By promoting a pro-inflammatory cytokine milieu, IRF5 facilitates the recruitment and activation of additional immune cells, including T cells, further exacerbating mucosal damage [19,20].

In addition, IRF5 regulates type I interferon pathways, which influence intestinal immune responses and the epithelial barrier function, potentially affecting epithelial integrity. Indirectly, IRF5-driven inflammatory mediators may compromise the barrier function, increasing intestinal permeability and perpetuating colitis [14].

It is important to note that the role of IRF5 in UC may vary between individuals and disease stages, and ongoing research continues to elucidate its precise contribution and potential as a therapeutic target.

Calprotectin is a reliable inflammatory marker that can be measured in the stool of IBD patients. Elevated levels (>50 µg/g stool) serve as a useful parameter for assessing the degree of inflammation [39].

In our study, the levels of calprotectin in the stool of our UC patients correlate with the number of positive cells in the biopsies of the colon. The correlation between elevated calprotectin levels and IRF5 expression strengthens the hypothesis that IRF5 could serve as an inflammatory marker in UC.

The interaction between IRF5 and PYK2, a non-receptor protein-tyrosine kinase, further elucidates the regulatory mechanisms at play. PYK2’s role as an upstream regulator of IRF5, facilitating its phosphorylation and subsequent activation, suggests a critical pathway through which IRF5 promotes the transcription of pro-inflammatory cytokines. This mechanistic insight opens avenues for therapeutic strategies targeting this pathway [27,34,40].

Defactinib, an orally bioavailable small-molecule focal adhesion kinase (FAK) inhibitor, has shown potential antiangiogenic and antineoplastic activities. It inhibits PYK2, and its effectiveness in tumor therapy has been examined in numerous studies. Clinically, defactinib is well tolerated, with minimal drug–drug interactions and consistent bioavailability independent of food intake. Its principal toxicities include low-grade nausea, fatigue, headache, and isolated, unconjugated, reversible Gilbert-type issues [41,42,43].

Defactinib is currently being investigated in many oncology studies and appears to show good efficacy in the treatment of, for example, small-cell lung cancer or pancreatic carcinoma [44].

A recent study showed a clear mechanistic interaction between PYK2 and IRF5 in macrophages, and in addition, an acceptable toxicological profile of the PYK2 inhibitor defactinib. The data also suggested a positive effect of defactinib on intestinal inflammation. The authors proposed that defactinib is an attractive molecule for repurposing to treat patients with UC or other inflammatory conditions, such as arthritis, acute lung injury, or atherosclerosis, in which IRF5 function in macrophages has been intimately linked to pathogenicity [34].

Numerous studies have confirmed that IRF5 plays an important role in the pathogenesis of IBD [5,8,10]. However, while the potential promise of defactinib as an IBD therapeutic has been identified, its effectiveness in IBD has yet to be adequately investigated.

In conclusion, our study establishes a significant correlation between IRF5 expression and disease activity in UC, proposing IRF5 as a novel marker for therapeutic intervention.

A limitation of this study is that only a small number of patients (thirty UC patients) and only eight healthy controls could be examined. Nevertheless, we were able to analyze a substantial number of biopsies overall, since biopsies were taken from all sections of the colon for each patient, which also exhibited varying stages of inflammation.

Further research is warranted to explore the exact mechanisms by which IRF5 contributes to the pathogenesis of UC, but it is also assumed to be involved in CD, so it is necessary to evaluate the therapeutic potential of targeting this pathway in clinical settings.

## Figures and Tables

**Figure 1 biomedicines-13-02251-f001:**
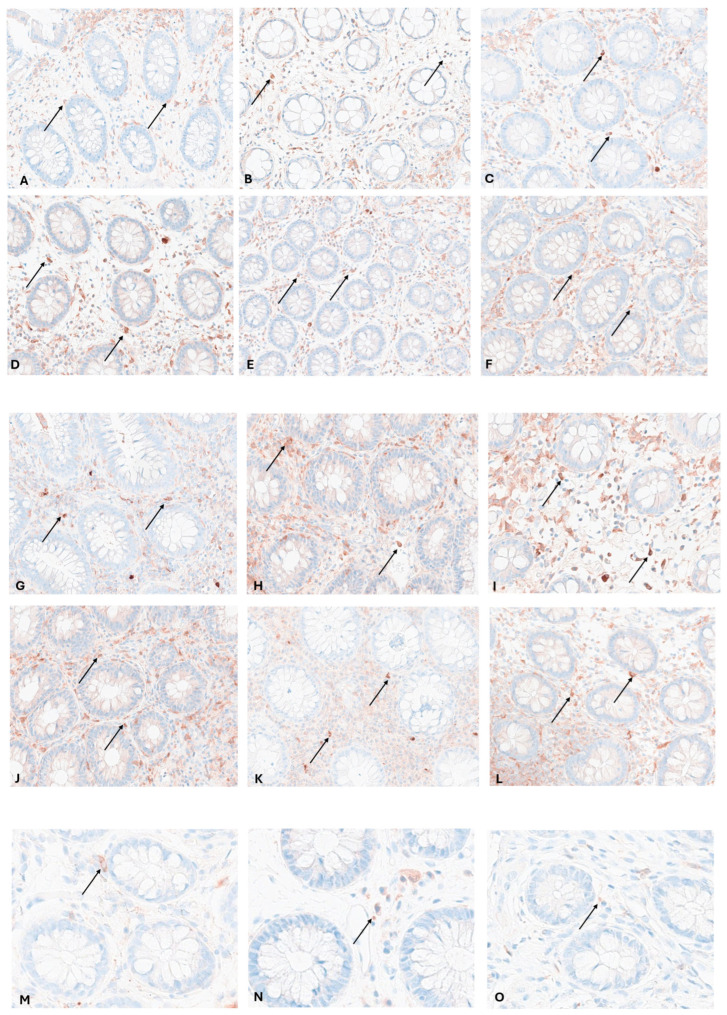
Immunostainings of bioptic samples from patients with UC. (**A**): Ascending colon in remission; (**B**): sigmoid colon in remission; (**C**): rectum in remission; (**D**): ascending colon with inflammation, Mayo Score 1-2; (**E**): descending colon with inflammation, Mayo Score 2; (**F**): sigmoid colon with inflammation, Mayo Score 2; (**G**): rectum with inflammation, Mayo Score 2; (**H**): colon transversum with inflammation, Mayo Score 3; (**I**): ascending colon with inflammation, Mayo Score 3; (**J**): sigmoid colon with inflammation, Mayo Score 3; (**K**): descending colon with inflammation, Mayo Score 2; (**L**): rectum with inflammation, Mayo Score 3; (**M**): caecum from healthy control; (**N**): descending colon from healthy control; (**O**)**:** rectum from healthy control. Arrows indicate IRF5+ cells (macrophage-like cells).

**Figure 2 biomedicines-13-02251-f002:**
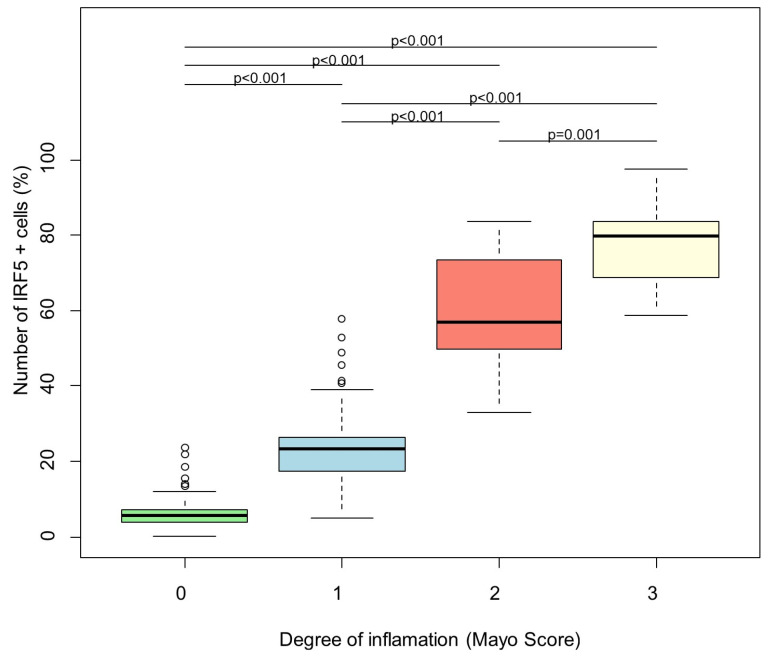
Linear regression analysis of number of IRF5 + cells and degree of inflammation. The degree of inflammation is a significant predictor for the number of IRF5 + cells (global *p* < 0.001). Bonferroni corrected *p*-values of post hoc pairwise comparisons between the inflammation degrees are shown on the figure. Green: Mayo Score 0, turquoise: Mayo Score 1, red: Mayo Score 2, yellow: Mayo Score 3.

**Figure 3 biomedicines-13-02251-f003:**
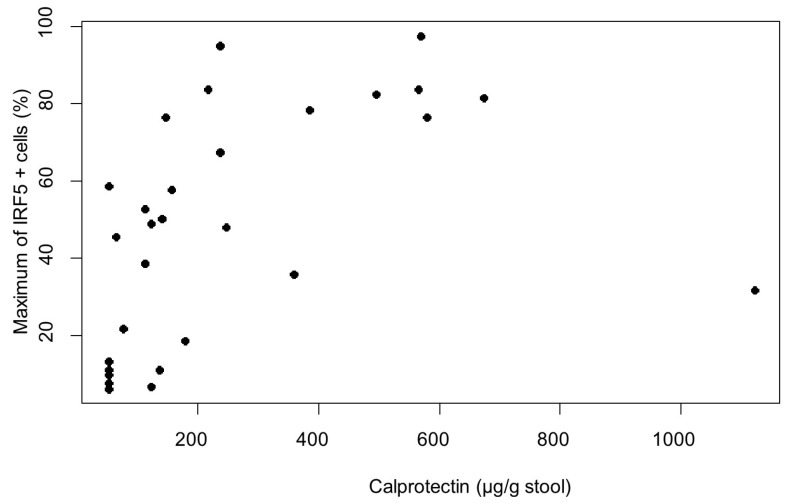
Linear regression analysis of number of IRF5 + cells and level of fecal calprotectin.

**Table 1 biomedicines-13-02251-t001:** Patient characteristics.

	Ulcerative Colitis
N (female)	30 (19)
Age (years), mean ± SD	42.7 ± 13.0
Disease localization	
Ulcerative Colitis, n (%)	
E1 proctitis	3 (10)
E2 left-sided colitis	9 (30)
E3 pancolitis	9 (30)
Remission	9 (30)
Medication, n (%)	
No treatment	0 (0)
5-ASA	8 (26.6)
Corticosteroids	2 (6.6)
Azathioprin	1 (3.3)
Sulfasalazine	1 (3.3)
Vedolizumab	9 (30)
TNF-alpha inhibitors	8 (26.6)
JAK inhibitors	1 (3.3)
Laboratory markers, median (min–max)	
Calprotectin [µg/mg]	252.76 (50–1122)

## Data Availability

The data that support the findings of this study are not openly available due to reasons of sensitivity and are available from the corresponding author upon reasonable request. Data are located in controlled access data storage at the University Hospital Frankfurt.

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
