# Peer review of "Interferon Regulator Factor 5: A Novel Inflammatory Marker and Promising Therapeutic Target in Ulcerative Colitis"

_biomedicines, 2025, doi:10.3390/biomedicines13092251_

Round 1
Reviewer 1 Report
Comments and Suggestions for Authors
This manuscript investigates the role of IRF5 in the pathogenesis of inflammatory bowel disease (IBD), with a particular focus on ulcerative colitis (UC). The authors performed immunohistochemical analyses of colon biopsies from UC patients and healthy controls, correlating IRF5 expression with clinical disease activity and fecal calprotectin levels. The study suggests that IRF5 may serve as a novel biomarker for UC and highlights its therapeutic potential, particularly in the context of emerging PYK2/IRF5 pathway inhibitors.
The manuscript describes IRF5+ "macrophage-like cells," but this conclusion is based solely on morphology in immunohistochemistry. Co-staining with established macrophage markers (e.g., CD68, CD163) would strengthen the claim. At minimum, limitations should be acknowledged.
Only 8 healthy controls were included, which is relatively small compared to the UC cohort. The manuscript should clarify whether control biopsies were taken from macroscopically and histologically normal mucosa. Some sentences in the introduction and discussion are repetitive (e.g., IRF5 role in autoimmunity mentioned multiple times). Consider tightening for conciseness.
This is a promising and well-structured manuscript addressing a clinically relevant question. However, concerns regarding sample size and integration of clinical confounders should be addressed before the work can be accepted.
Author Response
Thank you very much for taking the time to review this manuscript. Please find the detailed responses below and the corresponding corrections highlighted in the re-submitted files.
Comments 1: The manuscript describes IRF5+ "macrophage-like cells," but this conclusion is based solely on morphology in immunohistochemistry. Co-staining with established macrophage markers (e.g., CD68, CD163) would strengthen the claim. At minimum, limitations should be acknowledged.
Response 1: The Reviewer is correct in pointing out that macrophages can be detected immunohistochemically quiete well but detecting macrophages is a basic detection in histopathology and even 1st year residents learn how to correctly identify them. We are very confident that we correctly identified it. A double stain is technically impossible due to the fact that IRF5 and CD68 are both cytoplasmatic stains.
Comments 2: Only 8 healthy controls were included, which is relatively small compared to the UC cohort. The manuscript should clarify whether control biopsies were taken from macroscopically and histologically normal mucosa. Some sentences in the introduction and discussion are repetitive (e.g., IRF5 role in autoimmunity mentioned multiple times). Consider tightening for conciseness.
Response 2: Thank you for pointing this out. I agree with this comment. Therefore, I have made appropriate changes and added a limitation to the study.
Reviewer 2 Report
Comments and Suggestions for Authors
The current study is on a topic of general interest to the readers of the journal, where IBD is a severe health problem with high global rates of prevalence. Additionally, the article provides valuable insights into the role of IRF5 in UC and its potential as a biomarker and therapeutic target.
The study is novel as it addresses the role of IRF5 in ulcerative colitis (UC), which is a hot topic due to the need for novel biomarkers and therapeutic targets in inflammatory bowel disease (IBD). The authors presented a comprehensive methodology where the study employs systematic colon biopsies, immunostaining, and statistical analyses to evaluate IRF5 expression, thus ensuring robust data collection and analysis. The study demonstrates a significant correlation between IRF5 expression and disease activity in UC, suggesting IRF5 as a potential biomarker and therapeutic target. In addition, it underscores the potential of targeting the IRF5-PYK2 pathway with inhibitors like defactinib, opening avenues for new therapeutic strategies.
However, the study has a main limitation of using a small sample size, as the study includes only 30 UC patients and 8 healthy controls, which may limit the generalizability of the findings. This should be clearly stated in the limitations of the current study.
Author Response
Thank you very much for taking the time to review this manuscript. Please find the detailed responses below and the corresponding revisions highlighted in the re-submitted files.
Comments 1: However, the study has a main limitation of using a small sample size, as the study includes only 30 UC patients and 8 healthy controls, which may limit the generalizability of the findings. This should be clearly stated in the limitations of the current study.
Response 1: Thank you for pointing this out. I agree with this comment. I have therefore added this limitation to the manuscript.
Reviewer 3 Report
Comments and Suggestions for Authors
This study investigates the expression of interferon regulatory factor 5 (IRF5) in ulcerative colitis (UC) and its correlation with disease activity and inflammatory markers, proposing IRF5 as a potential biomarker and therapeutic target. The topic is interesting, however, there are some concerns which need to be addressed before publication.
- In the abstract, the background description is overly detailed, while the methods section is too brief. The methods should at least include key information such as sample size and specific statistical methods.
- While a correlation between IRF5 expression and inflammation is demonstrated, the functional mechanisms through which IRF5 contributes to UC (e.g., modulation of cytokine profiles, macrophage polarization, or interaction with microbial signals) remain unexplored. The authors should tone down mechanistic claims or clearly state that these require future validation.
- The discussion dedicates significant space to Defactinib and its role in oncology, which exceeds the direct scope of this study. It is recommended to focus more closely on the role of IRF5 in UC pathogenesis and clinical relevance.
- Both the potential of IRF5 as a biomarker for IBD and the role of Defactinib in alleviating intestinal inflammation (Nat Commun. 2021 Nov 18;12(1):6702. doi: 10.1038/s41467-021-27038-5) by targeting IRF5 have been previously reported. Please ensure that the Discussion clearly highlights the novel contributions of this study beyond these existing findings.
- The statistically significant differences in Figure 2 should be directly marked on the figure and explained in the figure legend.
Author Response
Thank you very much for taking the time to review this manuscript. Please find the detailed responses below and the corresponding corrections highlighted in the re-submitted files.
We used the editing service to improve the English language of the manuscript.
Comments 1: In the abstract, the background description is overly detailed, while the methods section is too brief. The methods should at least include key information such as sample size and specific statistical methods.
Response 1: Thank you for your comment. I agree with you and have added more detail about the methodology in the abstract.
Comments 2: While a correlation between IRF5 expression and inflammation is demonstrated, the functional mechanisms through which IRF5 contributes to UC (e.g., modulation of cytokine profiles, macrophage polarization, or interaction with microbial signals) remain unexplored. The authors should tone down mechanistic claims or clearly state that these require future validation.
Response 2: I agree with you on this point too and have added more details about the functional mechanisms through which IRF5 contributes to UC to the manuscript.
Comments 3: The discussion dedicates significant space to Defactinib and its role in oncology, which exceeds the direct scope of this study. It is recommended to focus more closely on the role of IRF5 in UC pathogenesis and clinical relevance.
Response 3: Based on your comment, I focused the discussion more specifically on the effect of IRF5 in the pathogenesis of UC. Nevertheless the connection to Defactinib remains very exciting, as it opens up new therapeutic options in the future.
Comments 4: Both the potential of IRF5 as a biomarker for IBD and the role of Defactinib in alleviating intestinal inflammation (Nat Commun. 2021 Nov 18;12(1):6702. doi: 10.1038/s41467-021-27038-5) by targeting IRF5 have been previously reported. Please ensure that the Discussion clearly highlights the novel contributions of this study beyond these existing findings.
Response 4: In this study, we aimed to demonstrate the importance of IRF5 in the pathogenesis of UC and provide initial evidence that IRF5 should be considered as a potential biomarker in the future. The cited study on defactinib confirms this approach, as IRF5 may be considered not only as a biomarker but also as a therapeutic target.
Comments 5: The statistically significant differences in Figure 2 should be directly marked on the figure and explained in the figure legend.
Response 5: We performed posthoc pairwise comparisons for the different degrees and marked the Bonferroni corrected p-values on the figure.